# Insights on Gut and Skin Wound Microbiome in Stranded Indo-Pacific Finless Porpoise (*Neophocaena phocaenoides*)

**DOI:** 10.3390/microorganisms10071295

**Published:** 2022-06-27

**Authors:** Chengzhang Li, Huiying Xie, Yajing Sun, Ying Zeng, Ziyao Tian, Xiaohan Chen, Edmond Sanganyado, Jianqing Lin, Liangliang Yang, Ping Li, Bo Liang, Wenhua Liu

**Affiliations:** 1Guangdong Provincial Key Laboratory of Marine Biotechnology, Institute of Marine Science, Shantou University, Shantou 515063, China; 20czli@stu.edu.cn (C.L.); 19hyxie@stu.edu.cn (H.X.); 14yjsun@stu.edu.cn (Y.S.); 14yzeng@stu.edu.cn (Y.Z.); 21zytian@stu.edu.cn (Z.T.); xh_chen@stu.edu.cn (X.C.); linjianqing@stu.edu.cn (J.L.); llyang@stu.edu.cn (L.Y.); liping@stu.edu.cn (P.L.); 2Southern Marine Science and Engineering Guangdong Laboratory (Guangzhou), Guangzhou 511458, China; 3Department of Applied Sciences, Northumbria University, Newcastle upon Tyne NE1 8ST, UK; esang001@ucr.edu

**Keywords:** gut microbiome, Indo-Pacific finless porpoise, skin wound, pathogens

## Abstract

The gut microbiome is a unique marker for cetaceans’ health status, and the microbiome composition of their skin wounds can indicate a potential infection from their habitat. Our study provides the first comparative analysis of the microbial communities from gut regions and skin wounds of an individual Indo-Pacific finless porpoise (*Neophocaena phocaenoides*). Microbial richness increased from the foregut to the hindgut with variation in the composition of microbes. *Fusobacteria* (67.51% ± 5.10%), *Firmicutes* (22.00% ± 2.60%), and *Proteobacteria* (10.47% ± 5.49%) were the dominant phyla in the gastrointestinal tract, while Proteobacteria (76.11% ± 0.54%), *Firmicutes* (22.00% ± 2.60%), and *Bacteroidetes* (10.13% ± 0.49%) were the dominant phyla in the skin wounds. The genera *Photobacterium*, *Actinobacillus*, *Vibrio*, *Erysipelothrix*, *Tenacibaculum*, and *Psychrobacter*, considered potential pathogens for mammals, were identified in the gut and skin wounds of the stranded Indo-Pacific finless porpoise. A comparison of the gut microbiome in the Indo-Pacific finless porpoise and other cetaceans revealed a possible species-specific gut microbiome in the Indo-Pacific finless porpoise. There was a significant difference between the skin wound microbiomes in terrestrial and marine mammals, probably due to habitat-specific differences. Our results show potential species specificity in the microbiome structure and a potential threat posed by environmental pathogens to cetaceans.

## 1. Introduction

The mammalian microbiome is a complex assembly of microorganisms vastly distributed in different anatomical niches of mammals. It is vital in mammalian health maintenance and disease defense [1]. Microbiome composition and structure are often organ-specific and linked to the physiological functions of the organs. The microbiome changes dynamically with host health status; minor disturbances in the microbiome may lead to its malfunction or subsequent sickness of the host. For example, *Proteobacteria* is one of the predominant microbiota constituents in the jejunum and is responsible for digestion, where *Proteobacteria* accounts for 2% of the total microbiome in the colon [2]. Dynamic changes in the richness of the microbiome, including *Proteobacteria*, *Fusobacteria*, and *Bacteroidetes*, were observed in morbidly obese patients [3], indicating a direct or indirect relationship between the microbiome and health status.

Additionally, other factors, including species [4,5], evolution [6], habitation [7], and dietary change [8], may also disturb the spatial and temporal composition of the microbiome. Consequently, cetaceans possess unique microbiomes in different anatomical niches, for example, the gastrointestinal tract and skin [9,10]. A previous study showed that a high similarity of function and higher-level taxonomy of the gastrointestinal tract microbiome were found between baleen whales and terrestrial herbivores; however, the protein catabolism and essential amino acid synthesis pathway in baleen whale microbiomes more closely resembled terrestrial carnivores, indicating that the diversity, structure, and function of the mammalian gut microbiome were mainly shaped by diet adaptation [10]. Moreover, differences in microbiome diversity and structure between different anatomical niches and species were found in a previous study. According to previous publications, *Firmicutes, Bacteroidetes, Gammaproteobacteria*, and *Fusobacteria* are the most abundant phyla in the midgut of the Indo-Pacific humpback dolphin—which inhabits the Pearl River Estuary of China—and short-finned pilot whales stranded in Hainan, China; however, a higher relative abundance of *Fusobacteria* was found in the stranded short-finned pilot whales than in the Indo-Pacific humpback dolphin [5,11]. For the gastrointestinal tract microbiome of the narrow-ridged finless porpoise, previous research found that the most abundant phylum was *Gammaproteobacteria* in the foregut, hindgut, and feces [12]. Both Indo-Pacific humpback dolphins and short-finned pilot whales belong to the family Delphinidae. In contrast, the narrow-ridged finless porpoise belongs to the family Phocoenidae, thus indicating that the gastrointestinal tract microbiome may be species-specific and related to evolutionary changes. In addition to the evolution-related changes and differences in species, habitations may also contribute to the differences in the microbiome of cetaceans, as different microbiome compositions were identified within captive and free-ranging common bottlenose dolphins, as well as common bottlenose dolphins living in different aquaria [7,13]. Due to complicated factors contributing to the dynamic changes in the microbiome of cetaceans, more individuals from different species should be included to better understand the microbiome in cetaceans.

Compared to the microbiome in the gastrointestinal tract, the microbiome inhabiting the skin may have a greater interaction with the mammals’ habitation/living environment [14]. Although marine mammals are constantly exposed to seawater microbiota [15,16,17], they also host their own unique microbial community, which is significantly different from the habitats on their skin [18,19,20]. The skin microbiome plays an important role in maintaining skin barrier function and the immune system while also preventing pathogen invasion. It is also important for wound healing [21,22,23,24]. A previous study demonstrated that *Staphylococcus* and *Pseudomonadaceae* are the most predominant bacteria in chronic human skin wounds [25,26,27]. Biofilms generated from *Staphylococcus aureus* suppress skin wound healing by inducing keratinocyte cell apoptosis or increasing cytokine expression in skin fibroblasts [28,29,30]. In contrast, *Staphylococcus epidermidis* was found to accelerate human skin wound healing via immune regulation [31,32], and both *Staphylococcus aureus* and *Staphylococcus epidermidis* protected the host from invasion by other pathogens through stimulation of the expression of antimicrobial peptides from human keratinocytes [33]. Previous studies have demonstrated the composition and structure of different species for the cetacean skin microbiome [19,20,28,29,34]. However, microbiome composition, structure, and function in cetacean skin wounds and their implications for cetacean health remain unclear.

Currently, more than 80 different species or subspecies of cetaceans have been found on earth; even though there is growing interest in research on the microbiome of cetaceans, little is known about the unique gastrointestinal tract microbiome or skin microbiome of cetaceans due to legal and ethical constraints. In this study, we investigated the microbiome in the gastrointestinal tract (foregut, midgut, hindgut) and skin wounds of a stranded Indo-Pacific finless porpoise (*Neophocaena phocaenoides*) from Eastern Guangdong waters, China. We aimed to investigate the diversity and composition of the Indo-Pacific intestinal microbial community. Additionally, we provide baseline information on the cetacean wound microbiome.

## 2. Materials and Methods

### 2.1. Sample Collection

Microbiome samples were collected from an Indo-Pacific finless porpoise (*Neophocaena phocaenoides*) that was washed onto Nan’ao Island (Shantou, Guangdong, China) in December 2019. The carcass was examined for external lesions, and entanglement marks were found (Figure 1). The status of the carcass was evaluated using international guidelines and was shown to be in good health at the time of death. It was designated code 2 (i.e., it was a freshly dead carcass with a time of death less than 24 h as suggested by its normal appearance, the absence of swelling or discernable smell, little or no sloughing on the skin and little or no change in eye or mucus membranes). Then the carcass was weighed and measured, and age was estimated by body length [30]. For nutritional state evaluation, the blubber thickness of two anatomical locations was measured (sternal and caudoventral to the dorsal fin) [35]. Appendix A shows age, sex, weight, and nutritional state information. Organs were examined visually, and no significant lesions were found.

For the intestinal tract, microbiome samples from the foregut, midgut, and hindgut contents were collected as described in a previous study [11,12]. Microbiome samples from skin wounds were collected with sterile swabs (Figure 1). All samples were collected in triplicate and stored at −80 °C until analysis. Sample collection was approved by the Regulations of the People’s Republic of China for the Implementation of Wild Aquatic Animal Protection and followed the guidelines and legal requirements in China.

### 2.2. Genomic DNA Extraction

Gastrointestinal tract samples were extracted with the MagPure Stool DNA KF kit B (Magen, Guangzhou, China) following the manufacturer’s instructions. For skin wound samples, collected samples were placed in TENS buffer (5 M sodium chloride, 10% SDS, Triton X-1000, Tris-HCl, EDTA), 10% SDS, and 20 mg/mL proteinase K and then incubated overnight at 55 °C. Proteins were removed by phenol/chloroform/isoamyl alcohol extractions, and DNA was precipitated with isopropanol. After washing in 75% ethanol twice, extracted DNA was resuspended in TE buffer. DNA from both gastrointestinal tract samples and skin wound samples was examined with a Qubit Fluorometer by using a Qubit dsDNA BR Assay kit (Invitrogen, Waltham, MA, USA), and the quality was further checked by gel electrophoresis.

### 2.3. Library Construction and Sequencing

Variable regions V3–V4 of the bacterial 16S rRNA gene were amplified with degenerate PCR primers 341F (5′-ACTCCTACGGGAGGCAGCAG-3′) and 806R (5′-GGACTACHVGGGTWTCTAAT-3′). Both forward and reverse primers were tagged with Illumina adapter, pad, and linker sequences. PCR enrichment was performed in a 50 μL reaction containing 30 ng template, fusion PCR primer, and PCR master mix. The PCR products were purified with Ampure XP beads and eluted in elution buffer. Libraries were qualified by an Agilent 2100 bioanalyzer (Agilent, Santa Clara, CA, USA). The validated libraries were sequenced on an Illumina MiSeq platform (BGI, Shenzhen, China) following the standard pipelines of Illumina.

### 2.4. Sequencing Data Processing

After sequencing, adaptors were removed, low-quality raw reads were filtered, and then paired-end reads were merged into tags by FLASH (v1.2.11) [36]. The chimera sequences were removed after comparison with the Gold database using UCHIME (v4.2.40) [37], and tags were clustered into OTUs with a cutoff value of 97% using Usearch software (v7.0.1090) [38]. The representative OTU sequences were classified with Ribosomal Database Project (RDP) Classifier v.2.2 with a minimum confidence threshold of 0.6 [39].

### 2.5. Microbiome Diversity and Structure Comparison

Microbial diversity and community structure were assessed using alpha diversity estimated with observed species (Sob), Chao1, Shannon, and Simpson indices, while beta diversity was estimated using the Weight UniFrac index. The microbial diversity estimators were evaluated with MOTHUR and QIIME [40,41]. Principal coordinate analysis, based on weighted UniFrac distances, was performed to assess the bacterial taxonomic composition of the gastrointestinal tract microbiome with and without the skin wound microbiome. Analysis of similarities (ANOSIM) was conducted to estimate the potential difference in the microbiome community structure from different regions of the Indo-Pacific finless porpoise.

### 2.6. Statistical Analyses

Statistical analyses were carried out with R software (Vienna, Austria; http://www.R-project.org/ (accessed on 23 April 2021)). The Wilcoxon rank-sum test was applied for pairwise comparisons of the bacterial diversity between groups, and *p* < 0.05 was considered statistically significant. The Benjamin–Hochberg procedure was used to control the false discovery rate due to multiple testing.

## 3. Results

### 3.1. Subsection

In total, 818,397 clean reads were obtained from 12 samples, including microbiome samples from the foregut, midgut, hindgut, and skin wounds, with a mean of 68,199 sequences per sample (Appendix A). After binning the 673,809 tags, according to >97% sequence identity, 211 operational taxonomic units (OTUs) were obtained (Appendix A). According to the rarefaction curve, the current sequencing depth was sufficient for taxa identification (Appendix A).

The observed species (Sobs) and Chao1 indices were applied for richness analysis for all four groups. As expected, the highest richness was found in skin wounds. No significant difference was found among the richness of the microbiomes of different gastrointestinal tract regions (*p* ≥ 0.1, *Wilcoxon rank-sum test*). However, the microbial richness index (both Sobs and Chao1) increased along the intestinal tract (foregut < midgut < hindgut) (Figure 2A,B).

For microbial species evenness, even though the highest Shannon index and lowest Simpson index were found in the skin microbiome, the midgut microbiome had the highest Shannon index and lowest Simpson index. However, there was no significant difference among the microbiomes from different anatomical niches, which indicated no significant difference in the species evenness of the four groups (Figure 2C,D).

The overall microbiome composition of the microbiome from the gastrointestinal tract and skin wounds was compared using principal coordinate analysis, nonmetric multidimensional scaling analysis, and analysis of similarities (ANOSIM). For the microbiome from the gastrointestinal tract, subjects between the two groups were separated, indicating a different structure of microbiome composition (Figure 3A,C). For analysis of similarities, R = 1 and *p* = 0.003 were found among the gastrointestinal tract microbiomes, which indicated variation in microbiome composition in different regions of the Indo-Pacific finless porpoise (Figure 3E). All results from the principal coordinate analysis, nonmetric multidimensional scaling analysis, and ANOSIM showed significant differences in the microbial community structure between the gastrointestinal tract microbiome and the skin wound microbiome (Figure 3B,D,F). This was probably due to the different microenvironments between the gastrointestinal tract and the skin wounds.

### 3.2. Microbial Taxonomic Profiles

The relative abundances of the bacterial phyla, orders, and genera of the microbiome of the gastrointestinal tract and skin wounds are shown in Figure 4. Thirteen phyla, including *Acidobacteria*, *Actinobacteria*, *Armatimonadetes*, *Bacteroidetes*, *Candidatus_Saccharibacteria*, *Chlamydiae*, *Cyanobacteria*, *Deinococcus thermmus*, *Firmicutes*, *Fusobacteria*, *Proteobacteria*, and *Verrucomicrobia*, were found in all samples. For the microbiome of the gastrointestinal tract, the top three most abundant phyla were *Fusobacteria* (67.51% ± 5.10%), *Firmicutes* (22.00% ± 2.60%), and *Proteobacteria* (10.47% ± 5.49%). For the relative abundance in skin wounds, the dominant microbial phyla were *Proteobacteria* (76.11% ± 0.54%), *Firmicutes* (12.00% ± 2.60%), and *Bacteroidetes* (10.13% ± 0.49%) (Figure 3A). The microbiome in the gastrointestinal tract was predominantly composed of *Fusobacteriales* (58.80–75.49%)*, Clostridiales* (17.66–25.61%)*,* and *Vibrionales* (5.14–18.34%). In contrast, the microbiome in the skin wounds was predominantly composed of *Vibrionales* (54.24–56.34%), *Erysipelotrichales* (11.97–13.32%), *Campylobacterales* (10.84–11.52%), *Flavobacteriales* (6.32–7.30%), *Bacteroidales* (3.08–3.46%), and *Pseudomonadales* (2.65–3.18%) (Figure 3B). The relative abundance of the order *Vibrionales* in the midgut was higher than that in the foregut and hindgut, but no statistical significance was found (Wilcoxon rank-sum test, *p* > 0.05). Interestingly, Vibrionales increased and then decreased from the foregut through to the midgut and hindgut; the opposite occurred with *Fusobacteria*.

At the genus level, 101 genera were detected from twelve samples, and different taxa dominated each sample. For gastrointestinal tract samples, *Cetobacterium* (67.51% ± 5.10%), *Clostridium_sensu_stricto* (18.69% ± 2.52%), *Photobacterium* (10.29% ± 5.52%), and *Clostridium_XI* (2.91% ± 0.56%) were the dominant genera. For the microbiome in skin wounds, *Vibrio* (50.36% ± 1.13%), *Erysipelothrix* (12.79% ± 0.73%), *Arcobacter* (11.22% ± 0.35%), *Photobacterium* (5.26% ± 0.09%), *Fulvibacter* (4.49% ± 0.43%), Psychrobacter (2.69% ± 0.21%), *Porphyromonas* (2.15% ± 0.20%), and *Tenacibaculum* (2.13% ± 0.09%) accounted for more than 90% of the genera. The microbiome in the skin wounds contained more genera than the microbiome from the gastrointestinal tract (Figure 4C).

### 3.3. Identification of Key Microbes in Each Anatomical Region

Linear discriminant analysis effect size (LEfSe) was used to compare bacterial abundances to identify the key bacteria (biomarkers) in different gastrointestinal tract regions. The key bacterial taxa driving the difference in the microbiome among different regions of the gastrointestinal tract are shown in Figure 5. For the microbiome from the gastrointestinal tract samples, key bacteria were only found in the midgut and hindgut but not the foregut (Figure 5). Within the midgut, one discriminant genus, *Photobacterium*, was identified. In addition, the high relative abundances of the phylum Proteobacteria, class *Gammaproteobacteria*, order *Vibrionales*, and family *Vibrionaceae* were significantly higher in the midgut than in the foregut and hindgut (Appendix A). In contrast, two discriminant genera were identified in the hindgut microbiome, i.e., *Actinobacillus* and *Pasteurellaceae* (Figure 5). Meanwhile, the abundances of the order *Pasteurellales* and the family *Pasteurellaceae* were higher in the hindgut than in the foregut and midgut (Appendix A).

## 4. Discussion

The microbiome is not only a critical factor for host health maintenance and disease progression but also an indicator for health assessment. However, relatively few studies have investigated the composition and structure of the microbiome in the cetacean gastrointestinal tract and skin wounds. In this study, the microbiome from the foregut, midgut, hindgut, and skin wounds of an Indo-Pacific finless porpoise was studied for the first time. We found different microbiome compositions between different gastrointestinal tract regions and further identified potential pathogens from skin wounds.

### 4.1. Microbiome of Gastrointestinal Tract of Indo-Pacific Finless Porpoise

The microbiome’s bacterial richness, evenness, and potential function were studied and compared among the gastrointestinal tract and skin wound samples. The results show that the richness and evenness of the microbiomes from different regions of the gastrointestinal tract were similar. The alpha diversity of bacterial communities in the present study was similar to that in previous studies on the Indo-Pacific humpback dolphin [11]. However, a higher Chao index and observed richness were found in the microbiome from the gastrointestinal tract of *Globicephala macrorhynchus*. Similarly, the microbiome evenness results are consistent with previous studies on Indo-Pacific humpback dolphins [11], narrow-ridged finless porpoises [12], short-finned pilot whales [5], and common minke whales [42]. The bacterial richness of the gastrointestinal tract microbiome of the Indo-Pacific finless porpoise in this study is comparable to the findings in the Indo-Pacific humpback dolphin [11] and the narrow-ridged finless porpoise [12], but lower than the bacterial richness found in the common minke whale [42]. The difference in microbiome composition may be individual- or species-specific. As reported in a previous study, the difference in health status or human care may contribute to the difference in the gut microbiome of cetaceans [4,43]. Moreover, common minke whales belong to the Balaenopteridae family, while both the Indo-Pacific finless porpoise and the narrow-ridged finless porpoise belong to the *Phocoenidae* family [30]. All Indo-Pacific finless porpoises, Indo-Pacific humpback dolphins, and narrow-ridged finless porpoises are coastal species, while short-finned pilot whales and common minke whales are offshore species [30]. Thus, the difference in the bacterial richness of the gastrointestinal tract of different marine mammals may be due not only to individual or species differences but also to habitations. However, due to the limited sample size (a single Indo-Pacific finless porpoise) included in this study, further analysis with a bigger sample size is required to better understand the microbiome of the Indo-Pacific finless porpoise.

According to the PCoA, NMDS, and ANOSIM analysis results, the microbial community compositions of the microbiomes from the foregut, midgut, and hindgut were clustered and significantly different from each other (Figure 3). Previous studies also found that the intestinal microbiome composition of cetaceans, such as Indo-Pacific humpbacks, short-finned pilot whales, and common minke whales, is significantly different among different regions of the gastrointestinal tract [5,11,42]. However, the microbiome structure of the narrow-ridged finless porpoise is similar along the gastrointestinal tract. It cannot be separated by NMDS, even though the narrow-ridged finless porpoise belongs to the same family, *Phocoenidae*, as the Indo-Pacific finless porpoise [12].

*Fusobacteria* (67.51% ± 5.10%), *Firmicutes* (22.00% ± 2.60%), and *Proteobacteria* (10.47% ± 5.49%) were the predominant phyla in the microbiome of the foregut, midgut, and hindgut, which is highly similar to the composition of the microbiome of short-finned pilot whales [5]. Unlike the composition of the microbiome of the Indo-Pacific finless porpoise in our study, a previous study showed that the predominant phyla in the Indo-Pacific humpback dolphin included not only *Fusobacteria* (14.44%), *Firmicutes* (47.05%), and *Proteobacteria* (14.82%) but also the phylum *Bacteroidetes* (23.63%) [11]. It has been reported that the phyla *Bacteroidetes* and *Firmicutes* were predominant in the microbiome of baleen whales, which is similar to the composition of the microbiome of terrestrial herbivores [10]. According to a previous report, the phylum *Bacteroidetes* plays an important role in polysaccharide degradation and is regarded as an important marker of health status in humans [44,45]. Disease progression may reduce the abundance of the phylum *Bacteroidetes* and increase the abundance of the phylum *Firmicutes* [45]. Thus, the reduced abundance of the phylum *Bacteroidetes* may also be due to the progression of disease in the Indo-Pacific finless porpoise stranded in Shantou.

The genus *Photobacterium* is vastly distributed in environmental media and marine animals [46,47]. According to previous studies, *Photobacterium* may cause gastroenteritis in humans [48]. *Photobacterium* infection normally occurs in immune-compromised individuals and is induced by consuming seafood, such as fish [46], which is the major prey of cetaceans. The discriminant genus *Photobacterium* was identified from the midgut and found in all gastrointestinal tract regions of the Indo-Pacific finless porpoise, indicating sickness (gastroenteritis) of the stranded Indo-Pacific finless porpoise. The genus *Photobacterium* was also found in other cetaceans, such as striped dolphins [49] and short-finned pilot whales [5], suggesting that the genus *Photobacterium* may not be a species-specific microbe in cetaceans. A previous study identified the genus *Photobacterium* from *L. vannamei* obtained from the local aquaculture farm of Shantou [50]. *Photobacterium* was also identified within the blowhole, mouth, tongue, and stomach of striped dolphins [49]. These results suggest that cetaceans may become infected by prey and highlight the importance of monitoring the local environmental microbiome for cetacean health management and conservation [51].

The genera *Haemophilus, Pasteurella,* and *Actinobacillus* belong to the family *Pasteurellaceae,* and both *Haemophilus* and *Pasteurella* were found in marine mammals, whereas the genus *Actinobacillus* was seldom found in marine mammals [52,53]. However, the genus *Actinobacillus* is a species-specific microbe of the Indo-Pacific finless porpoise which is still not clear. Meanwhile, the genus *Actinobacillus* will induce multiple organ infections or septicemia in terrestrial mammals [54,55], but the possible pathological mechanism in cetaceans is still not clear considering the special immune system of cetaceans [56].

### 4.2. Microbiome on the Skin Wounds of the Indo-Pacific Finless Porpoise

Similar to the human skin microbiome, the phyla *Proteobacteria*, *Firmicutes*, *Fusobacteria*, *Bacteroidetes*, and *Actinobacteria* are commonly found on the skin of cetaceans [9,20,29,32,57]. The composition of the skin microbiome is different between terrestrial and marine mammals, such as humans and cetaceans. The phylum *Actinobacteria* accounts for approximately 52% of the human skin microbiome but only accounts for approximately 8% of the skin microbiome of captive bottlenose dolphins and killer whales [20,58]. For the phyla *Proteobacteria* and *Bacteroidetes*, the percentage in the human skin microbiome is much lower than that in free-ranging humpback whales (16.5% vs. 42.2–61.1%, 6.3% vs. 30–82.2%) [20,28,29,57]. In contrast, the proportion of the phylum *Firmicutes* is much higher in humans than in cetaceans (24.4% vs. 7.3%) [20,59]. The difference in the skin microbiome between terrestrial and marine mammals may be due to the difference in the microbiome of the two habitats [6,10,18].

In this study, we identified a higher percentage of the phylum *Firmicutes* and a lower abundance of the phylum *Bacteroidetes* in skin wounds of the Indo-Pacific finless porpoise. In addition, a lower abundance of the phylum *Actinobacteria* was found in skin wounds of Indo-Pacific finless porpoises than in captive bottlenose dolphins and killer whales [20].

The genera *Vibrio* (50.36% ± 1.13%) and *Erysipelothrix* (12.79% ± 0.73%), considered potential cetacean pathogens, were identified from skin wounds of the stranded Indo-Pacific finless porpoise. *Vibrio owensii* accounted for 90% of the genus *Vibrio* in the skin wounds; however, its pathogenicity to mammals is still unclear. *Vibrio owensii* induces acute hepatopancreatic necrosis disease in invertebrates, such as shrimp and lobsters, but the possible infection cycle and pathogenicity in mammals are still unknown [60,61]. The high abundance of the genus *Vibrio* may be due to the indirect contact of the Indo-Pacific finless porpoise with invertebrates in the local habitat. *Erysipelothrix rhusiopathiae*, which is infectious to both livestock and humans [59,62], was the dominant species in the genus *Erysipelothrix* on the skin wounds of the Indo-Pacific finless porpoise. A previous study found that severe symptoms, including pulmonary edema, organ failure, and serosanguineous effusion, may occur in cetaceans due to the acute inflammatory reactions induced by *Erysipelothrix rhusiopathiae* [63]. The genera *Photobacterium*, *Tenacibaculum*, and *Psychrobacter* may also induce severe wound infection in animals [64,65,66]; however, the possible pathology in marine mammals is still unclear.

Taken together, we investigated the microbiome of a single Indo-Pacific finless porpoise from three different intestinal regions and skin wounds. Our results reveal the different compositions of the microbiome in different intestinal regions of the stranded Indo-Pacific finless porpoise. Meanwhile, we found a remarkable difference in the microbiome of infected skin wounds of the Indo-Pacific finless porpoise compared with that in terrestrial mammals, which may be due to the vast pathogens originating on land and the unique microbes from the sea within the habitations of cetaceans. However, due to the limitations of using only one individual porpoise in this study, whether the unique composition of the microbiome of Indo-Pacific finless porpoises is species-specific still remains unclear. Importantly, we found the possible pathogenic genera *Photobacterium*, *Actinobacillus*, *Vibrio*, *Erysipelothrix*, *Tenacibaculum*, and *Psychrobacter* from different anatomical niches of the Indo-Pacific finless porpoise. However, the pathogenesis of these pathogens in cetaceans is still not clear. Additionally, several studies have shown that the gut microbiome in mammals can change over time following death, which could have influenced the emergence and distribution of potential pathogens [67,68]. Thus, further studies of the microbiome, possible pathogens, and potential pathogenesis of pathogens in marine mammals should be conducted to better understand the mutual interactions of microbes and marine mammals and changes in microbiome structure following death.

## Figures and Tables

**Figure 1 microorganisms-10-01295-f001:**
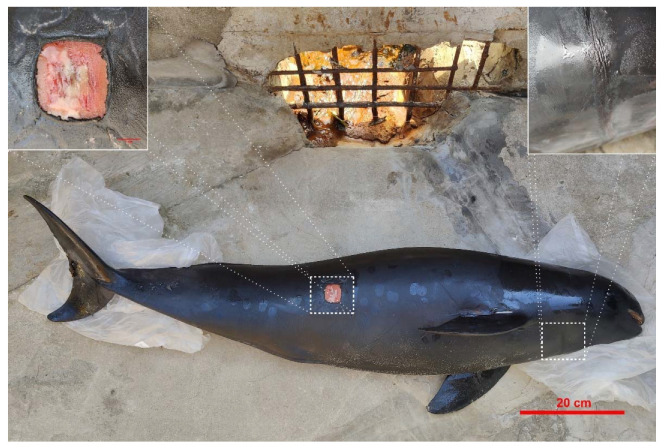
Image of the stranded Indo-Pacific finless porpoise.

**Figure 2 microorganisms-10-01295-f002:**
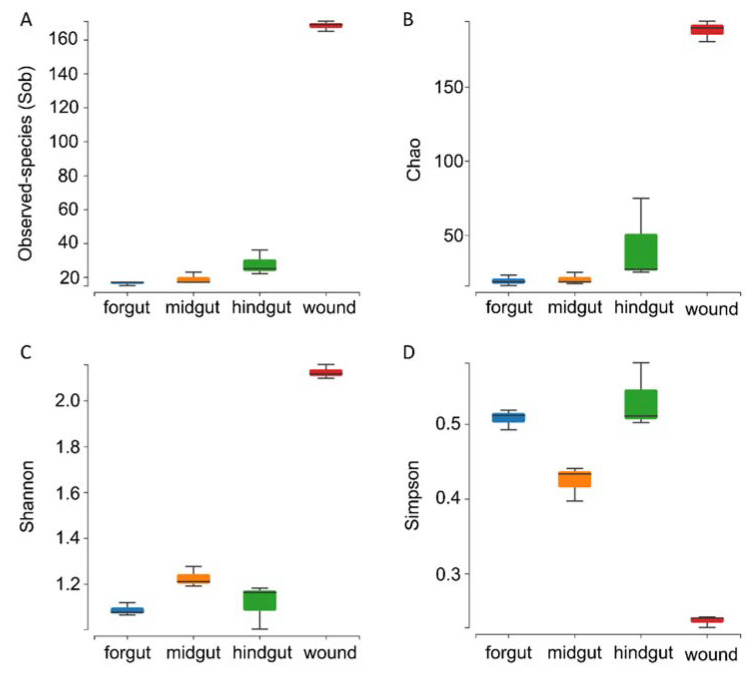
Alpha diversity, including observed species (Sob) (**A**), Chao1 (**B**), Shannon (**C**), and Simpson (**D**) indices, of the microbiome from the GI tract and skin wounds of an Indo-Pacific finless porpoise.

**Figure 3 microorganisms-10-01295-f003:**
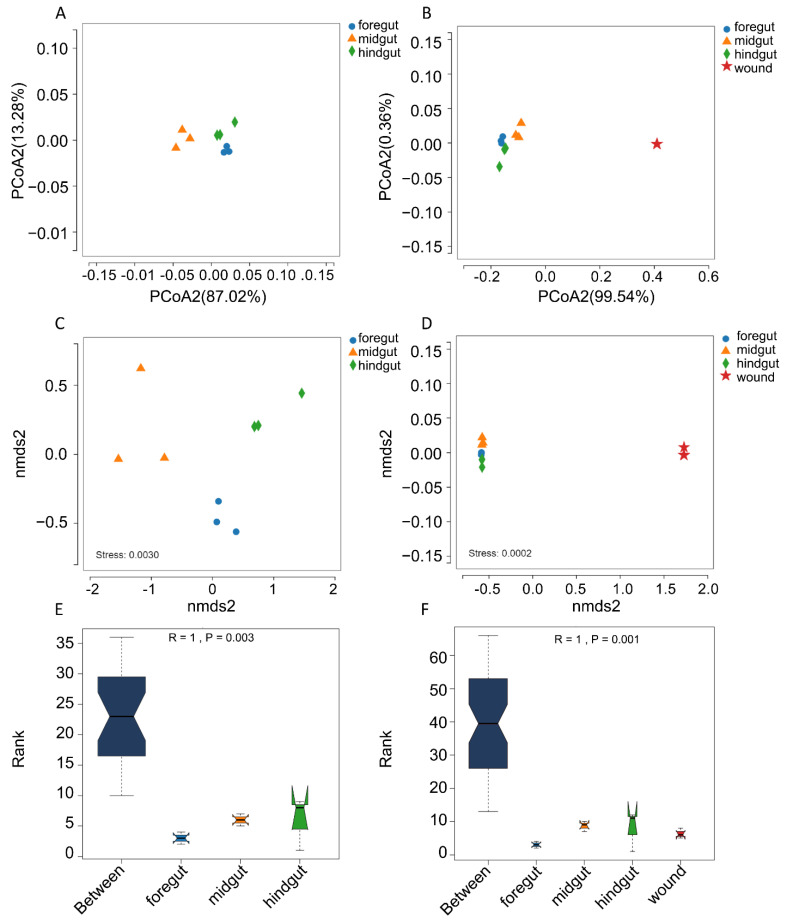
Principal coordinate analysis based on weighted UniFrac distances (**A**,**B**), nonmetric multidimensional scaling analysis (**C**,**D**) on the basis of the Bray–Curtis distance matrix, and analysis of similarities (ANOSIM) (**E**,**F**) of the microbiome from the GI tract (**A**,**C**,**E**) and wounds (**B**,**D**,**F**) of an Indo-Pacific finless porpoise.

**Figure 4 microorganisms-10-01295-f004:**
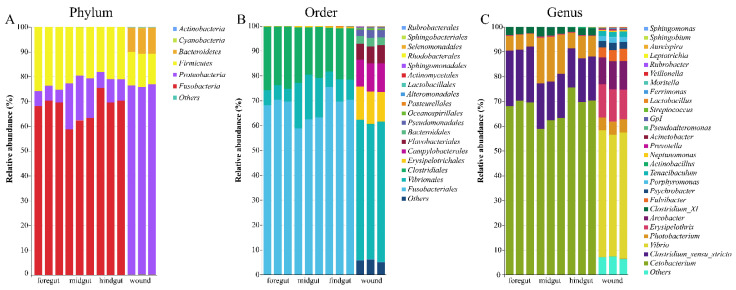
Relative abundance of the microbiome from the GI tract and skin wounds of an Indo-Pacific finless porpoise at the phylum level (**A**), order level (**B**), and genus level (**C**).

**Figure 5 microorganisms-10-01295-f005:**
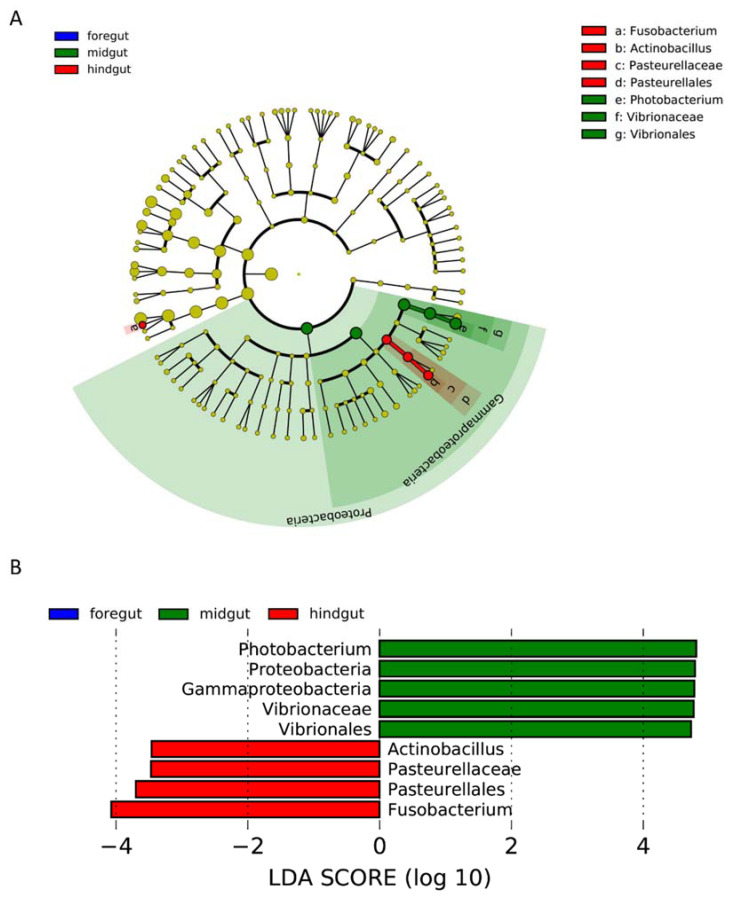
Linear discriminant analysis effect size (LEfSe) for key bacteria (biomarkers) identification for GI tract samples; results are shown in a cladogram (**A**) and histogram (**B**).

## Data Availability

Data were available from corresponding authors under reasonable request.

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
