# Peer review of "Insights on Gut and Skin Wound Microbiome in Stranded Indo-Pacific Finless Porpoise (Neophocaena phocaenoides)"

_microorganisms, 2022, doi:10.3390/microorganisms10071295_

Round 1
Reviewer 1 Report
Dear authors, a query for you.
While microbial signatures are highly individualised and multidimensional, and the content of the microbiome is generally highly correlated with an individual's diet, how do you explain the possibility of obtaining different results on the gut microbiome when the study sample is 2 or perhaps even 5 individuals?

Author Response
Reviewer 1 comments
While microbial signatures are highly individualised and multidimensional, and the content of the microbiome is generally highly correlated with an individual's diet, how do you explain the possibility of obtaining different results on the gut microbiome when the study sample is 2 or perhaps even 5 individuals?
Response:
Thank you for your comments. Previous studies in other mammals have shown that the microbiome is highly correlated to health status, species, evolution, habitat, anatomical niche, dietary change, etc. Our present study relied on opportunistic sampling of stranded animals, and provides the first insight on the differences between skin wound and gut microbiome. Sampling the cetacean we studied is restricted, hence to obtain more samples, waiting for more freshly dead strandings within the same regions may be required and this may take years or even decades. Our study provides an important baseline for future studies that will explore the role of gut microbiota as markers of cetacean health.
Reviewer 2 Report
In this study, the authors evaluate the microbiome of Indo-Pacific finless porpoises from three different intestinal regions and skin wounds.
After reading the manuscript, I have some concerns. Please read below.
In figure 2 it is extraordinary that there are no significant differences between the groups in view of the figure. Can you clarify/explain these results?
Why did you not use an ANOVA for the statistical analysis?
The figure 1 sample is not readable due to its size.
The conclusion section is reiterative after reading the extensive discussion it should be summarized and integrated into the discussion section.
In addition, the paragraph " Taken together, cetaceans may be threatened by pathogens originating from both terrestrial and marine environments, and the drastic difference in the microbiome between terrestrial and marine mammals suggested possible species- and habitation-specificity in microbiome composition on the skin wounds of cetaceans. Overall, the possibly unique pathogens and pathogenesis in marine mammals should be further studied" is already a conclusion.
In my opinion, the title of this manuscript is rather ambiguous.
The number of animals analyzed (n) is not indicated.
No characterization of the animals used: sex, weight, whether they all had the same basal health status or not, possible differences in age, etc.
Author Response
Reviewer 2 comments
In this study, the authors evaluate the microbiome of Indo-Pacific finless porpoises from three different intestinal regions and skin wounds.
After reading the manuscript, I have some concerns. Please read below.
In figure 2 it is extraordinary that there are no significant differences between the groups in view of the figure. Can you clarify/explain these results?
Response:
Thank you for your comments. We did identify that Sobs, Chao1, Shannon, and Simpson of wound microbiome were significantly different from gut microbiome (foregut, midgut, and hindgut) when using one-way ANOVA or t-test for the statistical analysis. For example, Fig 2A, Observed-species: Wound vs foregut, p<0.0001. However, when the Wilcoxon rank-sum test was used, Fig 2A, Observed-species: Wound vs foregut, p=0.0765, which did not fulfill the criteria we set in the ‘Materials and Methods’ section. Thus, we did not report any significant difference among these four groups.
Why did you not use an ANOVA for the statistical analysis?
Response:
Thank you for your comments. At the very beginning of data analysis, we tried to use an ANOVA method for the statistical analysis, but we found that the data did not pass the “Normality Tests”, such as the Anderson-Darling test or D'Agostino & Pearson test, due to limited n number.
Besides, we provided the statistical analysis results of alpha diversity by using one-way ANOVA as supplementary information for further reference.
The figure 1 sample is not readable due to its size.
Response:
Thank you for your comments. We have replaced the figure with a bigger size one.
The conclusion section is reiterative after reading the extensive discussion it should be summarized and integrated into the discussion section.
In addition, the paragraph " Taken together, cetaceans may be threatened by pathogens originating from both terrestrial and marine environments, and the drastic difference in the microbiome between terrestrial and marine mammals suggested possible species- and habitation-specificity in microbiome composition on the skin wounds of cetaceans. Overall, the possibly unique pathogens and pathogenesis in marine mammals should be further studied" is already a conclusion.
Response:
Thank you for your comments. We have deleted the following sentences in the manuscript and revised the conclusion.
In my opinion, the title of this manuscript is rather ambiguous.
Thank you for your comments. The title was revised as follows:
Novel Insights on Gut and Skin Wound Microbiome in Indo-Pacific Finless Porpoise
The number of animals analyzed (n) is not indicated.
Response:
Thank you for your comments. This study only studied a single fresh stranded Indo-Pacific finless porpoise. And we have revised and emphasized the sample size in subsection 2.1. Sample collection as follow now:
Microbiome samples were collected from only one Indo-Pacific finless porpoise (Neophocaena phocaenoides) stranded at the coast of Nan’ao Island (Guangdong, China) in December 2019.
No characterization of the animals used: sex, weight, whether they all had the same basal health status or not, possible differences in age, etc.
Response:
Thank you for your comments. Indo-Pacific finless porpoise inhabited in shallow, tropical to warm temperate waters, but still, quit far away from the land. Thus, the fresh stranding of Indo-Pacific finless porpoise was seldomly seen. In present study, we only have one Indo-Pacific finless porpoise was included, and the sex, weight, age, and blubber thickness for nutritional state assessment was added in supplementary table 1.
The following statement was added to subsection “2.1. Sample collection” for an explanation of the status of the stranded Finless porpoise:
The carcass was examined for external lesions, entanglement mark was found as shown in figure 1 (Fig 1). Then the carcass was weighed and measured; and age was estimated by body length [34]. Blubber thickness of two anatomical locations was measured (sternal and caudoventral to the dorsal fin) for nutritional state evaluation[35], information on age, sex, weight, and nutritional state were shown in supplementary table 1. Organs were examined visually, and no significant lesions was found.
Round 2
Reviewer 2 Report
My questions have been answered.
Author Response
Thank you so much for your valuable comments.